# Personal Aspects of Religiosity and Civic Engagement: The Mediating Role of Prayer

Małgorzata Tatala [1,*], Ryszard Klamut [2,*] and Celina Timoszyk-Tomczak [3,*]

1   Institute of Psychology, The John Paul II Catholic University of Lublin, 20-950 Lublin, Poland
2   Department of Humanities and Social Sciences, Rzeszow University of Technology, 35-959 Rzeszów, Poland
3   Institute of Psychology, University of Szczecin, 70-453 Szczecin, Poland
*   Correspondence: malgorzata.tatala@kul.pl (M.T.); rklamut@prz.edu.pl (R.K.);
    celina.timoszyk-tomczak@usz.edu.pl (C.T.-T.)

**Abstract:** The aim of the presented research was to test the relationship between basic beliefs about a religious object (Transcendence, God) operationalized by Hutsebaut and various forms of civic engagement. In discovering these relationships, the mediating role of prayer importance, understood as an indicator of the strength of commitment to the relationship with God, was taken into account. In the study conducted with 535 young adults, the following tools were used: Post-Critical Belief Scale, Civic Engagement Questionnaire and Prayer Importance Scale. The results showed that social activities were more strongly associated with beliefs accepting the existence of God, while political activities were more strongly linked with attitudes rejecting the existence of God. Prayer importance was shown to mediate the relationship between beliefs accepting the existence of God and social activities and to increase the strength of service-oriented activities.

**Keywords:** civic engagement; social and political activity; religiosity; post-critical beliefs; prayer; mediation





## 1. Introduction

Civic engagement refers to actions that are oriented toward the welfare of others as well as the community (Adler and Goggin 2005; Putnam and Campbell 2010). Thus, from a social perspective, it is a very beneficial phenomenon. It encompasses diverse activities, the aim of which is working for the common good to improve the quality of own and the community's life (Adler and Goggin 2005; Pancer 2015). Many authors have tried to classify such actions (e.g., Bekkers 2005; Zaff et al. 2008; Zalewska and Krzywosz-Rynkiewicz 2011). Klamut (2015) adopted two formal criteria as the basis for civic engagement classification, also listed in other categorizations, but not simultaneously: the area of involvement (service—power) and the level of community (individualism—collectivism (cf. Bekkers 2005)). Adopting both criteria makes it possible to distinguish and characterize four forms of civic engagement: Social Involvement, Social Participation, Individual Political Activity and Political Participation. Social Involvement is a type of individual activity aimed at helping other people in need (e.g., helping during natural disasters, participating in fundraising). Social Participation is also a type of activity directed toward helping others, but it is based on institutional capacity and is carried out communally, within the framework of organizations. The purpose of Individual Political Activity is to influence the way political scenes operate which involves increasing the understanding of socio-political reality and shaping it through informed participation in elections, demonstrations, signing petitions, etc., on an individual level. Finally, Political Participation is a collective action taken within organizations or political parties to influence the creation of new laws and to control their implementation (Klamut 2015). Previous research indicates that forms of civic engagement are linked to different political beliefs, as well as preferred values (Klamut 2013).

The welfare of others, both at the level of beliefs and action, is also an important aspect of religiosity (Chlewiński 1991; Zaff et al. 2008). Previous research studies found links between religiosity and civic engagement (cf. Caprara et al. 2018; Zaff et al. 2008; Pancer 2015). Pancer (2015) points out that reference to God is one of several important factors that trigger and direct civic action. The relationship with a superhuman and supra-material Transcendence helping to understand reality also determines how one relates to the world and to other people (Myers 2012; Anderson 2015; Pancer 2015).

Researchers have written about religious sources of beliefs on social reality and civic engagement, both in the social and political spheres (Jones-Correa and Leal 2001; Putnam and Campbell 2010; Zaff et al. 2008; Pancer 2015; Wallman Lundåsen 2022). It was shown that religious people are more involved in informal helping behaviors. Among these, neighborly help including simple housework, assistance in finding employment or lending a hand to the homeless is singled out (Putnam and Campbell 2010). Moreover, religious individuals donate more money and their free time to charity than non-religious people. They are more active volunteers and participants in their communities (Lam 2002; Loveland et al. 2005; Putnam and Campbell 2010). They are more likely to be a part of local organizations, be involved in working on projects aimed at solving problems in the community and hold office in such administrations. Religiosity is also associated with more frequent voting in local elections, participation in protests, marches or demonstrations and taking part in other public activities (such as parent council meetings) (Putnam and Campbell 2010; Lewis et al. 2013). Religiosity is related to political beliefs: more religious people are more likely to be characterized by a preference for right-wing and conservative ideologies (cf. Caprara et al. 2018). On the other hand, religiosity is associated with lesser support for civil liberties, such as issuing permits for meetings that promote unpopular views, intolerance and racism (Putnam and Campbell 2010).

Religiosity, similarly to civic engagement, is also complex in nature, which often causes difficulties in interpreting results obtained from research (Klamut et al. 2023). Bearing this complexity in mind makes it possible to define its role in motivational processes with greater accuracy (Anderson 2015; Szcześniak and Timoszyk-Tomczak 2020; Saroglou 2021). Studying the importance of religiosity in relation to civic activity, Lewis and colleagues (2013) made a distinction between behavioral and cognitive areas in religiosity. The first area, referred to as religious attendance, concerns the frequency of participation in services or other forms of worship and the nature of these activities. The second area, personal aspects of religiosity, involves the religious and theological beliefs and the cognitive framing constructed by religious knowledge, beliefs and experiences (Lewis et al. 2013; Anderson 2015).

Religious attendance mechanisms, based on involvement in religious services and other activities, are described most often in the literature on the subject (Verba et al. 1995; Jones-Correa and Leal 2001; Zaff et al. 2008; Putnam and Campbell 2010; Pancer 2015). A number of studies show a link between religious participation and pro-social activity (Lam 2002; Lewis et al. 2013; Pancer 2015). In the US, for example, those who regularly attended religious services were more likely to regularly participate in community service or volunteer (Wilson and Musick 1997) than those who were not members of religious organizations or communities (Lam 2002).

Religious attendance has also been connected with political participation. In a classic study on adult political activity in the US, Verba and colleagues (1995) showed that more frequent attendance at religious services was associated with a greater likelihood of participating in local and national elections, as well as engaging in a wide range of political activities, such as election campaigns, attending rallies or signing petitions, which are forms of direct citizen participation in the governance process (Jones-Correa and Leal 2001). As recognized by Pancer (2015), religious organizations provide resources that both initiate and sustain civic engagement. These organizations build a sense of community among those who take part in civic activities both inside and outside the church. This participatory type of motivation, however, stems not only from faith in God but also includes a social motif. It would be appropriate to emphasize the significant influence of clergy, leaders and

the fellowship of believers in shaping identification and a sense of belonging toward the religious group (Pancer 2015).

The second area explaining the importance of religiosity in civic engagement is the personal aspects of religiosity. It is expressed in a certain belief system, worldview or attitudes. The cognitive framing resulting from them (Lewis et al. 2013) provides specific criteria for understanding the world and taking action. In the area of beliefs, on the one hand, it is about the influence of religiosity on the internalization of altruism and selflessness, which can give rise to the need to act for the betterment of those in need. On the other hand, these are religious and theological beliefs that build a certain image of reality that goes beyond the material world. Beliefs about the existence of God and His nature, afterlife and salvation can foster pro-social behavior (Lewis et al. 2013; Saroglou 2021). In addition to directly impacting beliefs, religiosity influences civic engagement through the cognitive framework it creates. Religious people form a set of beliefs or attitudes that emphasize concern for others and an active contribution to improving the near and distant environment. This involvement includes not only helping those in need but also striving for social justice and fighting for economic and political rights in the community (cf. Myers 2012; Pancer 2015).

One of the personal aspects of religiosity is the basic belief regarding the existence and interpretation of a religious object (Transcendence, God) (Fontaine et al. 2005), also understood as an attitude toward religious interpretation (Wulff 1991; Szydłowski et al. 2021). Fundamental beliefs concern the elementary reference of a person to a religious object and, per Wulff (1991), are based on two criteria. The first involves believing in God: whether one includes Transcendence in their picture of the world or rejects God, focusing on the mortal world (inclusion vs. exclusion of Transcendence). The second signifies the way a person relates to religious content, interpreting it literally, analytically or symbolically (literal vs. symbolic interpretation). The combination of the indicated criteria makes it possible to distinguish four attitudes. Hutsebaut (1996) and Duriez et al. (2002) operationalized Wulff's theory (1991) in terms of post-critical beliefs, defining the attitudes as follows: Orthodoxy (per Wulff: literal affirmation)—literal inclusion of Transcendence; External Critique (per Wulff: literal disaffirmation)—literal exclusion of Transcendence; Relativism (per Wulff: reductive interpretation)—symbolic exclusion of Transcendence; and Second Naiveté (per Wulff: restorative interpretation)—symbolic inclusion of Transcendence.

Orthodoxy means recognizing the reality of a religious object and interpreting it literally, which can be exemplified by religious fundamentalism (Wulff 1991). External Critique is defined as a position denying the veracity of religious claims in the literal sense. This attitude usually refers to the exclusive recognition of claims based on scientific results and rational criteria, as is the case with anti-religious orientation or atheism. Relativism is an attitude that denies the reality of religious objects, transcendent references to God or religious practices. However, symbolic meanings of religious myths and rituals are accepted. This approach is characterized by exploration, rejecting doctrine and relativizing facts. Second Naiveté recognizes transcendent reality as real but avoids interpreting religious aspects explicitly and literally, rather seeking symbolic meanings (Wulff 1991; Hutsebaut 1996; Szydłowski et al. 2021). It is worth noting that in the presented work, we focus on the understanding of transcendence as seen by Wulff (1991) and Hutsebaut (1996), while the issue of transcendence is diversely analyzed within various sciences and theories. For example, in the context of spirituality, whether identified with religion or not, transcendence is oriented toward the sacred, or the sacred is treated broadly as a person, object, principle or concept transcendent of the "self" (Timoszyk-Tomczak and Bugajska 2016).

So far, attitudes toward religious interpretation have been used to explain political and social beliefs and behavior. Orthodoxy and Second Naiveté were found to correlate positively with cultural conservatism, and Relativism correlated negatively with economic conservatism. In contrast, racism was linked positively with Orthodoxy and External Critique (i.e., literal beliefs) and negatively with Relativism (Duriez and Hutsebaut 2000).

Subsequent studies involving the context of preferred values have shown that acceptance of Transcendence (Orthodoxy and Second Naiveté) is characteristic of those who favor tradition and conformity, benevolence and security. Those who reject Transcendence (External Critique and Relativism) opt for hedonism, stimulation and self-control. Literal interpretation of religious content is characteristic of the preference for values of power and security, while a symbolic interpretation is associated with universalism and benevolence (Fontaine et al. 2005; Śliwak and Zarzycka 2013). In the behavioral area, Schwadel (2005) noted the importance of religious literalism in engaging in civic activities. Those who interpreted the contents of the Bible more literally participated in church activities but were less likely to join non-church social organizations.

Prayer can also be considered an important personal aspect of religiosity, an indicator of the importance of religiosity in a person's life (Tatala and Wojtasiński 2021). Lewis and colleagues (2013) see it as a factor involved in the creation of cognitive framing. It occupies a central place in the life of Christians because it is realized through a personal relationship with God. Although by its nature it is a religious activity, it also has important psychological aspects, such as the subjective and personal, autonomous and authentic ability to enter into dialogue with God (Walesa 2005). The importance of the relationship with God in the individual sense expresses the strength of attachment to Him, while in the social context, it signifies caring and taking action for other people. Prayer can divert attention from personal matters and direct focus to the needs of others: loved ones, neighbors, but also the local or larger communities (Loveland et al. 2005). Studies have shown that people who prayed frequently and read religious texts were more likely to volunteer (Lam 2002). In addition, the frequency of prayer was related to helping behaviors toward known and unknown people, even after accounting for other religious and sociodemographic factors (Sharp 2019). Jankowski and Sandage (2014) showed that meditative prayer promotes the activation of positive emotions and pro-social behavior.

Prayer also has a role in increasing political awareness, which can lead to greater political involvement (Poloma and Gallup 1991). In contrast, it has not been found to significantly affect membership in political organizations (Putnam 2000).

The impact of prayer on civic participation also depends on the level of involvement in religious institutions. Higher contribution, particularly in volunteering in organizations focused on caring for others, enhances its impact. Individual prayer helps to identify with those who suffer and become involved in changing their lives (Ladd and Spilka 2002). Prayer can thus act as a mediator in the relationship between religious participation and social engagement. In scientific research on religiosity, cognitive and emotional processes were examined through prayer. However, its mediating function has been limited to testing its different types, rather than its importance, which could serve as an indicator of the commitment to a relationship with God (e.g., Walesa and Tatala 2020).

Most studies on the relationship between religiosity and social activity have included adults fully participating in civic life. However, there is a noticeable paucity of research among people in the period of emerging adulthood. Previous analyses (Walesa 2005) indicate the importance of distinguishing this developmental period due to the specificity of religiosity experienced then and participation in the community (Pancer 2015).

The period of emerging adulthood is defined by the following indicators: professional and financial independence, moving out of the family home, new place of residence, choosing a career, taking responsibility for oneself and for others, defining a path in life, finding a close social group and assuming civic responsibility (Arnett 2007). Young people develop in their chosen social roles, explore ideas for success and build a professional identity (Czerwińska-Jasiewicz and Wojciechowska 2011). This is also a time of confronting dreams or earlier visions of self with the realities of adult life. On the one hand, this is a period characterized by a mindset of taking sometimes risky action; on the other, heavy burdens can be felt then, due to accumulating commitments in many areas of life.

It is also a time when previous ideals and beliefs may change or weaken. Nowadays, values that can lead to success are preferred; work is treated as a means to material well-

being, happiness and satisfaction (Timoszyk-Tomczak 2010). Nevertheless, achieving psychological satisfaction expressed in experiencing love, family happiness and satisfying work turns out to be of most importance. In addition, the change of social roles involving a transition from dependence and subordination to independence and self-reliance is the most crucial developmental task. Adequate fulfillment of those developmental milestones provides a sense of happiness and ensures social approval which may inspire undertaking civic activities (Havighurst 1981). Religiosity provides a significant source of direction and motivation in young people (cf. Walesa 2005).

The study was carried out in one country, so it is worth pointing out some information on religiosity in Poland, also indicating the specifics of the studied group of young adults. A recent report, Church in Poland 2023, indicated an increase in secularization, the phenomenon proceeding differently in different social environments and age groups. It was noted primarily in large cities and among educated people, especially those aged 18–24. The results of the report also showed a slow decrease in the level of faith, with a relatively faster decline in religious practices (Tatala and Wojtasiński 2023). However, compared to other countries, the level of religiosity in Poland is still high.

In addition, the report (Kościół w Polsce 2023) indicated that 21% of respondents cited personal beliefs and thoughts as the reason for the lack of faith or indifference toward the Church, while 19% revealed discouragement toward the Church as an institution. At the same time, young people expected the Church to provide them with an experience of God (32.5%), support in crisis situations (29.7%) and arguments for the truthfulness of their professed faith (27%). It should be observed that the decreasing level of overall religiosity among the young is not always accompanied by a complete departure from the Church, but rather, other forms of religiosity appear.

The above-described dependencies between civic engagement, religiosity and prayer gave rise to more detailed hypotheses, which are presented below. The conducted research focused on analyzing emerging adulthood and was conducted with participants in the age range of 18 to 30.

## 2. Present Study

The aim of the presented research was to demonstrate what personal aspects of religiosity are predictors of different types of civic engagement in young people in the period of emerging adulthood. The hypothesized model includes (1) basic attitudes about the religious object (Transcendence, God), operationally referred to as post-critical beliefs (Hutsebaut 1996)—Orthodoxy, Second Naiveté, External Critique and Relativism—as the independent variables, (2) prayer importance as an indicator of the strength of commitment to the relationship with God as the mediator and (3) four forms of civic engagement—Social Involvement, Social Participation, Individual Political Activity and Political Participation— as the dependent variables.

The primary research question is finding what post-critical beliefs are associated with which forms of civic engagement. The second research question concerns the mediating role of prayer importance in the relationship between the studied variables. Previous research findings indicated that religious involvement plays a role in both pro-social activity and politics (Jones-Correa and Leal 2001; Lewis et al. 2013; Pancer 2015; Sharp 2019). However, the importance of religious beliefs, especially beliefs regarding reference to a religious object, has not been clearly defined.

Based on previous knowledge and the role played by religiosity in the life of a young person (cf. Walesa 2005), we posit that beliefs accepting the existence of Transcendence should indeed be associated with service-oriented activities. In contrast, beliefs rejecting God, indicating the non-acceptance of a Higher Power in one's view of the world, are associated with political activities. Notions other than religious ones, such as power and social influence, or values of hedonism, stimulation and self-control are significant here (Jones-Correa and Leal 2001; Lam 2002; Loveland et al. 2005; Pancer 2015; Sharp 2019).

Prayer is also likely to play an important role. We propose that it is positively linked with beliefs accepting Transcendence and negatively linked with beliefs rejecting Transcendence. As indicated by previous research, prayer is proposed to be a mediator in the relationship between religious beliefs and service-oriented activities (cf. Lam 2002; Loveland et al. 2005; Sharp 2019).

A review of the literature on the subject allowed us to formulate the following hypotheses:

**H1.** *Post-critical beliefs accepting Transcendence (Orthodoxy and Second Naiveté) correlate with social activities (service-oriented: Social Involvement and Social Participation).*

**H2.** *Post-critical beliefs rejecting Transcendence (External Critique and Relativism) correlate with political activities (Individual Political Activity and Political Participation).*

**H3.** *Prayer is positively related to post-critical beliefs accepting Transcendence and negatively related to post-critical beliefs rejecting Transcendence.*

**H4.** *Prayer acts as a mediator in the relationship between post-critical beliefs and service-oriented activities.*

**H5.** *The relationship between post-critical beliefs and service-oriented activities is mediated by prayer. The relationship between post-critical beliefs and political activities is direct.*

This hypothesis shows the relationships in a broader mediation model which takes into account the interdependencies between the independent and dependent variables studied.

## 3. Materials and Methods

### 3.1. Participants

This study included 535 adults between the ages of 18 and 30 ($M$ = 21.34; SD = 2.12), and 71.6% of the sample were women. The respondents resided in different areas of Poland, with the largest number from the Zachodniopomorskie, Podkarpackie and Lubelskie provinces. As their place of residence, the participants indicated a village (38.1%), a city of up to 20 thousand inhabitants (9.0%), a city of 20 to 100 thousand inhabitants (17.9%) and a city of more than 100 thousand inhabitants (34.6%).

The respondents were recruited with the snowball sampling technique, considering the criterion of age. The quantitative research was conducted with the distributed survey method. It began with the introduction performed by the researcher. After establishing amicable contact, the participants were informed of the purpose of the study, which was anonymous and voluntary in nature. Responses were provided on paper questionnaires and completed by hand.

### 3.2. Measures

Three research tools were used in the study: Post-Critical Belief Scale by Hutsebaut in the Polish adaptation by Bartczuk et al. (2011), Civic Engagement Questionnaire by Klamut (2015) and Prayer Importance Scale (PIS) by Tatala and Wojtasiński (2021).

#### 3.2.1. Post-Critical Belief Scale (PCBS)

The scale is used to measure cognitive references to Transcendence/God and ways of interpretation of the religious content. It consists of 33 items, and the responses are rated on a 7-point scale from 1 = completely disagree to 7 = completely agree. The results fall into a two-dimensional space. The first dimension indicates the extent to which people accept the existence of God or some other transcendent reality (inclusion vs. exclusion of Transcendence). The second dimension depicts how religious content is interpreted (literal vs symbolic interpretation). As a result, the participant obtains scores in four subscales:

Orthodoxy, Second Naiveté, External Critique and Relativism. Reliability of the scale in the Polish samples was found to be satisfactory and varied across studies: for External Critique, $\alpha$ = 0.84 to 0.89, for Orthodoxy, $\alpha$ = 0.50 to 0.73, for Relativism, $\alpha$ = 0.67 to 0.74, and for Second Naiveté, $\alpha$ = 0.56 to 0.71 (Bartczuk et al. 2011).

### 3.2.2. Civic Engagement Questionnaire (CEQ)

The tool to measure the level of involvement in various types of civic activities: Social Involvement (SI), Social Participation (SP), Individual Political Activity (IPA) and Political Participation (PP). The questionnaire consists of 17 items, and the responses are given on a 5-point scale, from 1 = definitely no to 5 = definitely yes. The scales are characterized by high validity and reliability, with Cronbach's alpha coefficients for SI ranging from $\alpha$ = 0.70 to 0.76, for SP, $\alpha$ = 0.74 to 0.76, for IPA, $\alpha$ = 0.65 to 0.77, and for PP, $\alpha$ = 0.75 to 0.82 (Klamut 2015).

### 3.2.3. Prayer Importance Scale (PIS)

The scale is used to assess the subjective importance of prayer in a person's life. It provides information about the individual level of religiosity. It consists of six items, and responses are scored on a 5-point scale from 1 = strongly disagree to 5 = strongly agree. The scale's internal consistency, measured by Cronbach's alpha coefficient, was $\alpha$ = 0.90 (Tatala and Wojtasiński 2021).

## 4. Results

Statistical analyses were performed using R 4.3.1 for Windows with package lavaan (Rosseel 2012). First, we verified the data to ensure that the parametric requirements were met, i.e., the multivariate normal distribution of the variables, the presence of outliers and the linearity and homoscedasticity of the distribution of interdependence (co-variance) of the variables. We carried out the preliminary analyses with Tidyverse (Wickham et al. 2019) and Psych (Revelle 2023) packages. Descriptives of the analyzed variables are presented below. We tested the assumption of a normal distribution with the statistics of skewness, kurtosis and the Shapiro–Wilk test (Morgan and Griego 1998) (Table 1).

**Table 1.** Descriptives.

| Scale | Variable | M | SD | Skewness | Kurtosis | W—Shapiro–Wilk Test |
|-------|----------|----|----|----------|----------|---------------------|
| | PCBS | | | | | |
| | Orthodoxy | 2.75 | 1.25 | 0.47 | −0.49 | 0.96 |
| | Second Naivete | 4.06 | 1.40 | −0.41 | −0.62 | 0.97 |
| | Relativism | 4.15 | 1.15 | −0.65 | 0.227 | 0.97 |
| | External Critique | 3.36 | 1.25 | 0.10 | −0.55 | 0.99 |
| | CEQ | | | | | |
| | Social Involvement | 2.87 | 0.94 | 0.01 | −0.67 | 0.98 |
| | Social Participation | 1.99 | 0.99 | 0.91 | 0.01 | 0.88 |
| | Individual Political Activity | 3.16 | 0.91 | −0.09 | −0.60 | 0.99 |
| | Political Participation | 1.33 | 0.66 | 2.73 | 8.10 | 0.57 |
| | PIS | | | | | |
| | Prayer | 2.57 | 1.36 | 0.15 | −1.42 | 0.88 |

Skewness and kurtosis scored less than ±2 with most variables, which allowed us to assume that they were within the acceptable limits of a normal distribution (Fidell and Tabachnick 2003). However, the variable of Political Participation PP was completely different from the normal distribution, which we took into account when analyzing and interpreting the results.

In order to verify hypotheses H1–H3, correlation analyses (Pearson's *r*) were conducted between post-critical beliefs, civic engagement and prayer importance (Table 2).

**Table 2.** Correlation coefficients between the analyzed variables.

| Scale | Variable | [1] | [2] | [3] | [4] | [5] | [6] | [7] | [8] | [9] |
|---|---|---|---|---|---|---|---|---|---|---|
| | | PCBS | | | | | | | | |
| [1] | Orthodoxy | | | | | | | | | |
| [2] | Second Naiveté | 0.66 * | | | | | | | | |
| [3] | Relativism | 0.01 | 0.37 * | | | | | | | |
| [4] | External Critique | −0.28 * | −0.31 * | 0.51 * | | | | | | |
| | | CEQ | | | | | | | | |
| [5] | Social Involvement | 0.26 * | 0.27 * | 0.06 | −0.06 | | | | | |
| [6] | Social Participation | 0.32 * | 0.32 * | −0.02 | −0.21 * | 0.52 * | | | | |
| [7] | Individual Political Activity | −0.06 | 0.08 | 0.21 * | 0.19 * | 0.19 * | 0.22 * | | | |
| [8] | Political Participation | 0.28 * | 0.02 | −0.10 | 0.08 | 0.21 * | 0.38 * | 0.16 * | | |
| | | PIS | | | | | | | | |
| [9] | Prayer | 0.67 * | 0.71 * | 0.01 | −0.51 * | 0.29 * | 0.39 * | 0.01 | 0.08 | |

* Minimal significant correlation with *p* = 0.05 is *r* = |0.09|.

The results showed that Social Involvement SI correlated significantly and positively with Orthodoxy O (*r* = 0.26) and Second Naiveté SN (*r* = 0.27). Similarly, Social Participation SP correlated positively with O (*r* = 0.32) and SN (*r* = 0.32). Additionally, it correlated negatively with External Critique EC (*r* = −0.21). Individual Political Activity IPA correlated significantly and positively with Relativism R (*r* = 0.21) and EC (*r* = 0.19), and Political Participation PP correlated positively with O (*r* = 0.28). Social Involvement SI (*r* = 0.29) and Social Participation SP (*r* = 0.39) were also positively associated with Prayer P. Significant correlations were observed between post-critical beliefs and prayer, and so, strong positive correlations were reported between O and P (*r* = 0.67) and SN and P (*r* = 0.71), negative correlations were found between EC and P (*r* = −0.51).

To verify hypothesis H4, we conducted the mediation analysis using the procedure proposed by Preacher and Hayes (2008), the asymptotically distribution free method (ADF) for SEM (Jaiswal and Niraj 2011). The results are shown in Table 3.

Analyzing the obtained results, we concluded that prayer is a mediator in the relationship between O and SI (*b* = 0.06; *p* = 0.01), and total mediation was observed (*b* = 0.09, *p* = 0.46). Prayer is also a mediator in the relationship between SN and SI (*b* = 0.06; *p* = 0.01), and total mediation was confirmed (*b* = 0.11, *p* = 0.09). Therefore, the results support hypothesis H4. In addition, prayer proved to be a mediator in the relationship between EC and SI (*b* = −0.05; *p* = 0.01). A partial suppression effect was found in this relationship (*b* = 0.14, *p* = 0.02), indicating that a lack of involvement in prayer stifles the effect of the relationship between EC and SI.

Prayer was a significant mediator in the relationship between O and SP (*b* = 0.07; *p* = < 0.001), the mediation effect was close to total (*b* = 0.04, *p* = 0.45), and the direct relationship between the two variables when controlling for prayer was significant at the accepted level of *α* = 0.05. Prayer was also a significant mediator in the relationship between SN and SP (*b* = 0.08; *p* = 0.01), and total mediation was confirmed (*b* = 0.09; *p* = 0.09). Involvement in prayer suppressed the relationship between EC and SP (*b* = −0.06; *p* = < 0.01). Suppression was complete (*b* = 0.03; *p* = 0.55). In the other cases, the presumed mediating effect was not observed.

**Table 3.** Mediation model.

| Model | | Effect | | | | | | | | | | | | |
|---|---|---|---|---|---|---|---|---|---|---|---|---|---|---|
| | | Indirect | | | | Direct | | | | Total | | | |
| | | *b* | *β* | SE | Z | *b* | *β* | SE | Z | *b* | *β* | SE | Z |
| SI | *O* | 0.06 | 0.07 | 0.03 | 3.04 * | 0.04 | 0.05 | 0.05 | 0.77 | 0.09 | 0.12 | 0.05 | 1.88 |
| | *SN* | 0.06 | 0.09 | 0.02 | 3.02 ** | 0.11 | 0.16 | 0.06 | 1.87 | 0.15 | 0.23 | 0.06 | 2.70 ** |
| | *R* | 0.01 | 0.00 | 0.01 | 0.08 | −0.08 | −0.09 | 0.06 | −1.33 | −0.07 | −0.09 | 0.06 | −1.30 |
| | *EC* | −0.05 | −0.07 | 0.02 | −2.96 ** | 0.13 | 0.17 | 0.05 | 2.56 ** | −0.02 | −0.03 | 0.05 | −0.40 |
| SP | *O* | 0.07 | 0.09 | 0.02 | 3.75 ** | 0.04 | 0.05 | 0.05 | 0.76 | 0.11 | 0.14 | 0.05 | 2.18 * |
| | *SN* | 0.08 | 0.12 | 0.02 | 3.74 ** | 0.09 | 0.13 | 0.06 | 1.61 | 0.18 | 0.28 | 0.06 | 3.24 ** |
| | *R* | 0.01 | 0.01 | 0.01 | 0.08 | −0.07 | −0.09 | 0.06 | −1.33 | −0.07 | −0.09 | 0.06 | −1.29 |
| | *EC* | −0.06 | −0.08 | 0.02 | 3.63 ** | 0.03 | 0.04 | 0.05 | 0.61 | 0.07 | 0.09 | 0.05 | 1.26 |
| IPA | *O* | 0.03 | 0.04 | 0.02 | 1.60 | −0.14 | −0.20 | 0.05 | −2.98 ** | −0.12 | −0.16 | 0.04 | −2.56 ** |
| | *SN* | 0.03 | 0.05 | 0.02 | 1.60 | 0.12 | 0.19 | 0.06 | 2.19 * | 0.15 | 0.24 | 0.05 | 2.93 ** |
| | *R* | 0.01 | 0.01 | 0.01 | 0.08 | 0.02 | 0.02 | 0.06 | 0.32 | 0.02 | 0.02 | 0.06 | 0.32 |
| | *EC* | −0.03 | −0.03 | 0.02 | −1.59 | 0.18 | 0.24 | 0.05 | 3.50 ** | 0.15 | 0.21 | 0.05 | 3.14 ** |
| PP | *O* | 0.01 | 0.01 | 0.01 | 0.32 | 0.17 | 0.41 | 0.03 | 6.49 ** | 0.18 | 0.42 | 0.03 | 7.05 ** |
| | *SN* | 0.01 | 0.01 | 0.01 | 0.32 | −0.04 | −0.10 | 0.03 | −1.26 | −0.04 | −0.09 | 0.03 | −1.22 |
| | *R* | 0.01 | 0.01 | 0.01 | 0.08 | −0.09 | −0.20 | 0.03 | −3.02 ** | −0.09 | −0.20 | 0.03 | −3.02 ** |
| | *EC* | 0.01 | −0.01 | 0.01 | −0.32 | 0.12 | 0.28 | 0.03 | 4.26 ** | 0.11 | 0.27 | 0.03 | 4.37 ** |

** $p < 0.01$; * $p < 0.05$. Note: SI—Social Involvement, SP—Social Participation, IPA—Individual Political Activity, PP—Political Participation; O—Orthodoxy, SN—Second Naiveté, R—Relativism, EC—External Critique.

SEM structural equation analysis was performed to verify hypothesis H5. The asymptotically distribution free method (ADF; Jaiswal and Niraj 2011), which does not require variables to be normally distributed, was applied due to the variable of PP. Figure 1 presents the structural equation model showing the relationship between post-critical beliefs and civic engagement, taking into account the mediating role of prayer importance.

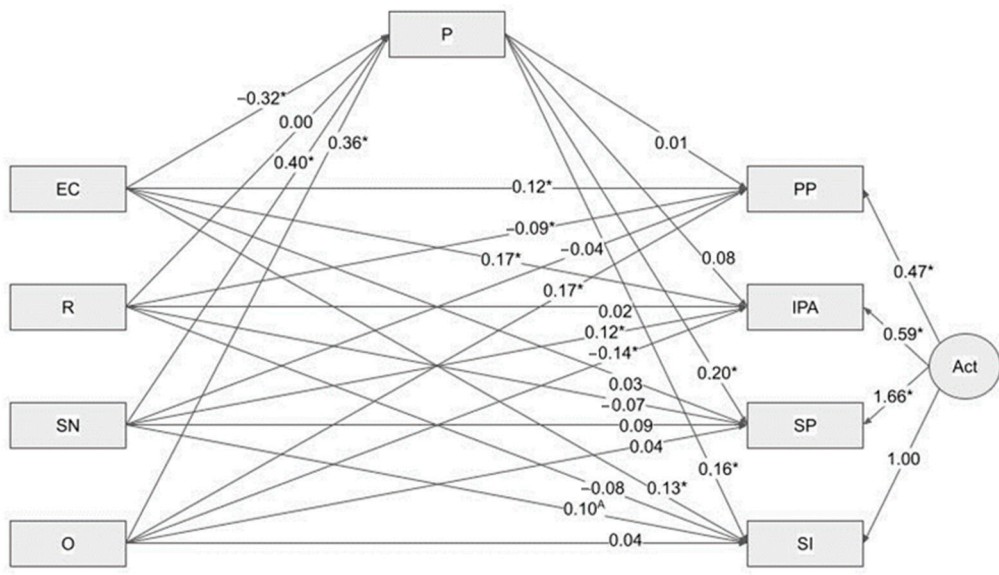

$X^2(2) = 8.547$; $p < 0.014$; $RMSEA = 0.08$; $CI_{90} = (0.03; 0.13)$; $CFI = 0.99$; $SRMR = 0.013$
* $p < 0.05$; ᴬ $< 0.07$

**Figure 1.** Structural equation model of the relationship between post-critical beliefs and civic engagement with the mediating role of prayer importance.

Fit indices were satisfactory; only the value of RMSEA = 0.08 was on the border of being acceptable (Kline 2011). This means that the adopted model fits the data. Post-critical beliefs

mediated by prayer importance were predictors of each form of civic engagement. The results obtained from the path analysis support hypotheses H4 and H5. The relationship between post-critical beliefs is mediated by prayer, and the relationship between post-critical beliefs and political activities is direct. Significant relationships between Orthodoxy O and Social Involvement SI, as well as between O and SP, were found, both mediated by P. The relationships between O and P ($\beta = 0.36$), P and SI ($\beta = 0.16$), P and SP ($\beta = 0.20$), O and SI ($\beta = 0.04$) and O and SP ($\beta = 0.04$) were found. Directly, O was negatively associated with IPA ($\beta = 0.14$) and positively associated with PP ($\beta = 0.17$). SN through P had a significant impact on SI and SP. The relationships between SN and P ($\beta = 0.40$), P and SI ($\beta = 0.16$), P and SP ($\beta = 0.20$), SN and SI ($\beta = 0.10$) and SN and SP ($\beta = 0.09$) were found. Directly SN was positively related to IPA ($\beta = 0.12$). R was directly and negatively related to PP ($\beta = -0.09$). EC was related to SI, directly ($\beta = 0.13$) and indirectly through P. In addition, EC was related to SP indirectly through P and directly to IPA ($\beta = 0.17$) and to PP ($\beta = 0.12$). The model also hypothesized the existence of the relationship between post-critical beliefs and prayer, and so the strongest positive links appeared between SN and P ($\beta = 0.40$) as well as O and P ($\beta = 0.36$). There was also a negative relationship between EC and P ($\beta = -0.32$). A significant positive relationship emerged between prayer and social activities and so between P and SI ($\beta = 0.16$) as well as P and SP ($\beta = 0.20$).

## 5. Discussion

The aim of the presented study was to determine the relationship between personal aspects of religion and civic engagement. The relations between cognitive constructs and taking action are not very strong and depend on various factors, including the concreteness of beliefs and the specificity of actions (cf. Aronson et al. 2012). This regularity also applies to beliefs about religion and activities related to religious practices and religious motivation expressed in terms of working for the benefit of others (Glazier 2020). Relationships between religion and civic engagement are manifold. Some forms of religious participation diminish, while others increase civic involvement (Lam 2002; Pancer 2015; Glazier 2020). Previous research indicates that regular attendance at religious services is associated with participation in the community or volunteer work (Wilson and Musick 1997). Factors such as social networks (Lewis et al. 2013; Pancer 2015), religious vs. secular schooling (Hill and Den Dulk 2013), socio-cultural aspects (Piatak 2023) and how the religious community interacts with social and political activities (Glazier 2020) are significant in the relationship between religiosity and civic engagement. Researchers are not in agreement on why religious people are more socially engaged or to what extent religious beliefs or activity is associated with political activity.

The study presented in this paper showed new aspects of those relationships, focusing on the role of personal aspects of religiosity (Lewis et al. 2013), particularly beliefs about the existence and nature of Transcendence and prayer. The primary objective of the research was to examine the relationship between religious beliefs and engagement in various forms of social and political action (H1–H3). The second objective was to test whether prayer serves as a mediator in the relationship between beliefs about religion and civic engagement (H4 and H5).

The conducted research confirmed the posited hypotheses expanding the knowledge on the analyzed variables. Beliefs regarding the existence of God were found to correlate with civic engagement. More specifically, beliefs accepting Transcendence were associated with social engagement (H1), while beliefs rejecting Transcendence were associated with political engagement (H2). Post-critical beliefs toward a religious object are cognitive-affective in nature and form the basis of a worldview (Wulff 1991; Hutsebaut 1996; Szydłowski et al. 2021). They are individual explanations of religion, which are motivationally significant. By definition, they can form the basis of behavioral decisions. The obtained results indicated that a sufficient factor for triggering service-oriented activity, whether undertaken individually or communally, is a belief in the existence of God. Such a statement seems to be too general, but this finding can provide an argument in the discussion on the role of religiosity

in taking social action. If a person includes God in their view of the world, they behave in accordance with this image (Jang et al. 2023). In addition, such activities are reinforced by the absence of beliefs rejecting God at the literal level (negative correlation with External Critique). In contrast, the dismissal of Transcendence at both literal and symbolic levels is linked to Individual Political Involvement. The reasons for this relationship can be found in the possibility of realizing values related to security and power (cf. Fontaine et al. 2005; Klamut 2013; Śliwak and Zarzycka 2013).

Post-critical beliefs accepting Transcendence correlated positively with prayer. In turn, beliefs rejecting Transcendence either negatively correlated with prayer, like External Critique, or did not correlate with prayer, like Relativism. For the most part, this supported hypothesis H3. If beliefs presuppose the existence of God, then it seems obvious to form a relationship with God, and prayer importance is an indicator of the degree of commitment to that relationship (Tatala and Wojtasiński 2021). If, on the other hand, beliefs reject God, as in External Critique, then prayer is also rejected. External Critique involves questioning the veracity of religious claims and accepting beliefs based on scientific results. It is associated with anti-religious orientation, which explains the negative correlation with prayer (Bartczuk et al. 2011). The lack of links between Relativism and prayer may be due to the nature of this belief, which denies the reality of a religious object but is associated with the need to give symbolic meaning to religious content. As can be seen, Relativism leads to other ways of resolving religious issues that have no place for prayer (Bartczuk et al. 2011; Szydłowski et al. 2021).

The results of this study supported hypothesis H4, indicating that the subjective meaning of prayer is an important mediator in the relationship between post-critical beliefs and service-oriented forms of civic activity. The results showed that the relationship of Orthodoxy and Second Naiveté with Social Involvement and Social Participation is fully mediated by prayer. This means that post-critical beliefs accepting Transcendence, especially when supported by prayer, translate into both individual and collective social activity. The results warrant a thesis that not only participation but also cognitive and emotional aspects are factors motivating social involvement (Lam 2002; Lewis et al. 2013). The results of the present study are consistent with those obtained by Jang and colleagues (2023), in which it was found that participants who believed in a Higher Power, practiced prayer, studied religious texts and took part in religious group activities, were more likely to feel a transcendent responsibility for their impact on other people and the environment to the Higher Power. Additionally, transcendent responsibility was linked to pro-social attitudes, and these attitudes were linked to social activism. Ladd and Spilka (2002) showed that prayer incorporates concern for others and increases civic and political awareness (Poloma and Gallup 1991). As a result, it can enhance self-efficacy through a sense of connection with God, which in turn translates into a person's sense of efficacy as a citizen (Poloma and Gallup 1991).

Hypothesis H5, postulating a mediating role of prayer in the relationship between post-critical beliefs and service-oriented activities, as well as the direct relationship between post-critical beliefs and political activity, was confirmed. The results take into account interdependencies between the studied independent and dependent variables in a broader mediation model. Religion affects social and political engagement differently through religious attendance and religious beliefs (Lewis et al. 2013). Previous research on religious beliefs connects it with social activism (Lewis et al. 2013; Saroglou 2021) as well as with political activism (Zaff et al. 2008; Putnam and Campbell 2010; Pancer 2015). Also, people with providential religious beliefs tend to have higher levels of social involvement but lower levels of political activity (Glazier 2020). Separate attitudes toward a religious object are also associated with separate civic activities.

In the conducted research, it was proposed that the primary criterion differentiating the importance of beliefs toward a religious object as a predictor of civic engagement is the acceptance/rejection of Transcendence. The obtained results indicated that this criterion does play a role. However, an interesting additional look at the importance of beliefs

rejecting Transcendence in taking political action revealed the role of the second criterion: the literal/symbolic reference to religious content. It distinguished between the ways the two beliefs rejecting the existence of God affect political activity. External Critique (literal rejection of Transcendence) correlated negatively with prayer, which suppressed the effect of the relationship between this attitude and social activities. In contrast, Relativism (symbolic rejection of Transcendence) was not related to prayer and directly affected political activities, with a negative relationship with Political Participation.

Analyses showed that in the relationship between post-critical attitudes and political activism, a criterion relating to how one interprets religious content is also important. Thus, not only is it significant whether someone believes but also how someone processes religious content (cf. Fontaine et al. 2005; Śliwak and Zarzycka 2013).

The relationship between attitudes accepting Transcendence and political activity further indicated that the greater the intensity of the Orthodox attitude, the less involvement in Individual Political Activity and the greater involvement in collective Political Participation. In contrast, the greater the level of Second Naiveté, the greater the Individual Political Activity, but there is no relationship with Political Participation. Indirectly, this result supports previous findings showing that more frequent participation in religious services is associated with a greater likelihood of taking part in local and national elections (Verba et al. 1995). It also increases involvement in a wide range of political activities such as elections, running for office, attending rallies and signing petitions (Jones-Correa and Leal 2001). The results indicate that social engagement occurs already as a result of activating cognitive categories—core beliefs about the existence of God—and then is further strengthened by the engagement in a relationship with God (prayer).

Post-critical beliefs rejecting Transcendence were the predictors of political activity. In the presented research, beliefs associated with External Critique, characterized by the denial of truth in religious claims, were linked with Individual Political Activity and Political Participation. More precisely, the higher the External Critique, the greater the Political Participation, both individual and collective. This can be related to the preferred values such as hedonism, stimulation, self-direction, security and power (Fontaine et al. 2005; Śliwak and Zarzycka 2013). The area of political activity is located in the area of power; hence, the attitude of External Critique may be the predictor of political activity (Klamut 2015). Indirectly linked to this are results indicating that a literal interpretation of the content of the Bible is associated with less frequent affiliation to non-church social organizations and more frequent membership in church-based organizations (Schwadel 2005).

Of interest is the attitude of Relativism rejecting Transcendence and treating religious content symbolically (Szydłowski et al. 2021). In the presented research, taking into account the relationship between the independent and dependent variables, Relativism was not associated with Individual Political Activity and slightly reduced the level of Political Participation. Individuals with this attitude generally do not engage in political activity.

The presented results confirm previous interpretations emphasizing that religion, also in the context of beliefs, can be one of the factors initiating and sustaining civic engagement (Pancer 2015; Lewis et al. 2013). In addition, prayer importance was a mediator in the relationship between post-critical beliefs accepting Transcendence and Social Involvement and Social Participation. As for the links between beliefs and political activity, beliefs related to the literal interpretation of religious content (External Critique) were the most significant direct predictors of political involvement.

Taking into account the specificity of the sample, it is worth noting that faith, prayer and activity undertaken for the benefit of others may be associated with religious authenticity characteristic of young adulthood (Walesa 2005). At this stage of life, religiosity is based on truthfulness, originality, sincerity and following one's ideals (Walesa 2005). Beliefs, feelings, decisions, relationships with others, practices and experiences become significant, especially when they originate from the young person and are not the expression of conforming to the environment or certain patterns of behavior prevailing in it. Religious authenticity requires a certain radicalism, which, on the one hand, is a challenge but, on

the other, may be attractive to a young person seeking unambiguity and clarity (Walesa and Tatala 2020).

The presented research results can be interpreted from much broader perspectives, including philosophical, sociological, political or historical ones. An interesting reference can be made to Gauchet's (2021) analysis on the profound transformation of the human social universe. In his view, it takes place on the foundation of religion but reverses its original logic. The author highlights the paradox associated with the development of human political and psychological autonomy. The growth of this autonomy results from changes in religious consciousness, which is based on the surge in divine power and its increasing distance from human activity. Perhaps such changes are the attitudes represented by the subjects of the presented study, which could be the subject of further analyses.

Due to the novel nature of the presented research findings, the obtained results should be interpreted with a level of caution. The empirical data refer primarily to a group of Polish Catholics, representing the period of emerging adulthood. The results showing the relationships between variables studied in one age group are certainly worth comparing with other developmental periods. This would allow us to confirm and cross-check the specificity of functioning in the period of emerging adulthood. Additionally, it would be interesting to study the anticipation of religiosity, prayer and civic engagement in a perspective of 10 years' time, as well as at the end of life, since the measure of proper functioning in certain areas of life involves taking different time perspectives into account. The outlook of a decade represents a clear, significant distance from the present time. Considering the period at the end of life constitutes criterionally and existentially the most focal perspective for human existence (cf. Walesa and Tatala 2020). Further research also could focus on investigating the importance of different aspects of religiosity, e.g., the created image of God, in undertaking civic activity, rather than the general importance of accepting or rejecting Transcendence per se. Some studies indicate relationships between the image of God and empathy in adolescents (Francis et al. 2012), which may indicate the basis for the assumed relationships.

**Author Contributions:** Conceptualization, M.T., R.K. and C.T.-T.; methodology, M.T., R.K. and C.T.-T.; formal analysis, M.T., R.K. and C.T.-T.; investigation, M.T., R.K. and C.T.-T.; data curation, M.T., R.K. and C.T.-T.; writing—original draft preparation, M.T., R.K. and C.T.-T.; writing—review and editing, M.T., R.K. and C.T.-T.; visualization, M.T., R.K. and C.T.-T.; project administration, M.T., R.K. and C.T.-T. All authors have read and agreed to the published version of the manuscript.

**Funding:** This research received no external funding.

**Institutional Review Board Statement:** The study was conducted in accordance with the Declaration of Helsinki, and approved by the Ethics Committee of the Institute of Psychology of the University of Szczecin (protocol code: 13/2023, date of approval: 25 May 2023).

**Informed Consent Statement:** Informed consent was obtained from all subjects involved in the study.

**Data Availability Statement:** Data supporting reported results will be available in the Institutional Repository of the John Paul II Catholic University of Lublin at the link: https://hdl.handle.net/20.500.12153/6176, accessed on 25 January 2024.

**Conflicts of Interest:** The authors declare no conflict of interest.

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
