# Peer review of "Personal Aspects of Religiosity and Civic Engagement: The Mediating Role of Prayer"

_religions, doi:10.3390/rel15020192_

Round 1
Reviewer 1 Report
Comments and Suggestions for Authors
I very much enjoyed reading this interesting and well-conceived piece of research. Here are my recommendations for improvement.
1. Some of the prose in the piece is clunky, and the paper overuses passive voice. I would recommend the authors do an edit throughout for this.
2. I’m unclear on what the last sentence on p. 1 is conveying (“Previous research…), and it seems odd to stand in a paragraph alone. Can the authors elaborate?
3. The sentence on line 90 (“Religious attendance has also been contended…” ) seems to have an odd word choice. A typo?
4. When the authors address the relationship between religious beliefs, especially about the afterlife, and civic participation, I do think they should acknowledge where beliefs about the afterlife might cause a divestment from civic participation, such as in sects that self-segregate from society.
5. Line 133: I would caution the authors against equating Orthodoxy with fundamentalism. I think there are important nuances there, and differences between them.
6. Line 156: is “displaying” the right word here?
7. I recommend the authors work further on Hypothesis 5. First, I’m not sure why we would expect the relationship to be FULLY mediated by prayer when there are many other aspects of religious life and practice that might also act as mediators. Second, the hypothesis is wordy and doesn’t “read” like a hypothesis, but rather like exposition.
8. I recommend the authors explain the context of religiosity in Poland in the first part of the article. No mention of the context is made until the “Method” section, and many readers might not know how to interpret results from that location. I would also like to know more about their sampling frame, whether it was conducted in a university setting or somewhere else.
9. Page 10 seems to have many extra blank spaces, is this a typo? Also, stylistically it isn’t my preference to print each of the values in the text that the reader can visualize in the figure, but I would instead recommend calling attention to the most notable conclusions from the SEM model.
10. Did the authors consider conducting multivariate regressions where they could also control for gender and place of residence? I am sensitive to the fact that the sample size constrains the complexity of the model, but I would be interested to see any differences by gender and place of residence, at the very least, since these variables are mentioned in the beginning of the Method section.
Comments on the Quality of English LanguageN/A
Author Response
Reviewer 1
Answer: We would like to thank you for your review and comments, which helped us to improve the quality of the article.
Must be improved
Are the research design, questions, hypotheses and methods clearly stated?
I very much enjoyed reading this interesting and well-conceived piece of research. Here are my recommendations for improvement.
1. Some of the prose in the piece is clunky, and the paper overuses passive voice. I would recommend the authors do an edit throughout for this.
Answer: APA guidelines allow the use of the passive side. However, it can sometimes be overused. We have reviewed the entire manuscript and made changes to improve the quality of the text. Thank you for pointing out this issue.
I'm unclear on what the last sentence on p. 1 is conveying ("Previous research...), and it seems odd to stand in a paragraph alone. Can the authors elaborate?
Answer: It was an unnecessary emphasis put on one sentence. The content of this paragraph has been merged with the previous paragraph.
The sentence on line 90 ("Religious attendance has also been contended..." ) seems to have an odd word choice. A typo?
Answer: It indeed was a typo. It was replaced with the correct word: „connected”.
When the authors address the relationship between religious beliefs, especially about the afterlife, and civic participation, I do think they should acknowledge where beliefs about the afterlife might cause a divestment from civic participation, such as in sects that self-segregate from society.
Answer: Indeed, this may be the case. However, we are concerned with clarifying the content of religious beliefs, hence we referred to the general supra-material categories quoting external sources. We have refined this sentence.
Line 133: I would caution the authors against equating Orthodoxy with fundamentalism. I think there are important nuances there, and differences between them.
Answer: Now, Line: 135: Certainly, Orthodoxy and fundamentalism are not the same. Hence, we pointed to fundamentalism as the example provided by the author of the concept. For greater precision, we supplemented the text with a reference to the publication of Wulff (1991).
Line 156: is "displaying" the right word here?
Answer: Now, Line 163: It was an error, we have corrected the sentence. Thank you for bringing it to our attention.
I recommend the authors work further on Hypothesis 5. First, I'm not sure why we would expect the relationship to be FULLY mediated by prayer when there are many other aspects of religious life and practice that might also act as mediators. Second, the hypothesis is wordy and doesn't "read" like a hypothesis, but rather like exposition.
Answer: Thank you for this comment, we accept it. We have removed the term "fully". We also separated the last sentence from the hypothesis. It is meant to explain the hypothesis, so we moved it below.
I recommend the authors explain the context of religiosity in Poland in the first part of the article. No mention of the context is made until the "Method" section, and many readers might not know how to interpret results from that location. I would also like to know more about their sampling frame, whether it was conducted in a university setting or somewhere else.
Answer: We have included information on religiosity in Poland and the survey sampling technique.
Page 10 seems to have many extra blank spaces, is this a typo? Also, stylistically it isn't my preference to print each of the values in the text that the reader can visualize in the figure, but I would instead recommend calling attention to the most notable conclusions from the SEM model.
Answer: In the text, we included the standardized coefficients using the Latin symbol b [Beta], which did not display when the Palatino Linotype font was applied to the text. Hence, we retyped the symbol b [Beta]. We opted for not changing the presentation of the results, as they are in accordance with APA standards.
Did the authors consider conducting multivariate regressions where they could also control for gender and place of residence? I am sensitive to the fact that the sample size constrains the complexity of the model, but I would be interested to see any differences by gender and place of residence, at the very least, since these variables are mentioned in the beginning of the Method section.
Answer: In the presented article, we did not include the regression analysis. The volume of the article and the adopted framework of the analyses limit the possibilities of examining additional variables. Instead, we are in the process of preparing another article, which will include multivariate regression analyses.
Best regards,
Reviewer 2 Report
Comments and Suggestions for Authors
This is an interesting empirical research proposed to study the mediating role of importance of religious practice for civic engagement. Two main comments to the authors:
1) The article is submitted to the Special issue on Spirituality, so how do the authors link the measures of importance of prayer with the theoretical framework of spirituality research? In order to be published in this Special issue, the authors have to consider the link by articulating it in the text.
2) How would the authors consider their results taking into consideration the results of the research on religiosity and God images (see Francis and Pyke 2012,
This would be interesting to discuss.
Technical issues: there are mistakes with spelling "Indiwidual Political Activity" in the table. Please correct this.
Comments on the Quality of English Language
The article is well written in academic English
Author Response
Reviewer 2
Comments and Suggestions for Authors
Answer: We would like to thank you for your review and sharing inspiring comments.
This is an interesting empirical research proposed to study the mediating role of importance of religious practice for civic engagement. Two main comments to the authors:
1) The article is submitted to the Special issue on Spirituality, so how do the authors link the measures of importance of prayer with the theoretical framework of spirituality research? In order to be published in this Special issue, the authors have to consider the link by articulating it in the text.
Answer: We also tried to include the concept of spirituality (Line 147). The article was originally dedicated to another special issue, but the editors offered us this one. Introducing the context of spirituality to a greater extent would be difficult due to the volume of the text.
2) How would the authors consider their results taking into consideration the results of the research on religiosity and God images (see Francis and Pyke 2012,
DOI: 10.1080/13617672.2012.732810) This would be interesting to discuss.
Answer: Thank you for referring us to an important and interesting text, we have quoted it in lines 583-588.
Technical issues: there are mistakes with spelling "Indiwidual Political Activity" in the table. Please correct this.
Answer: This has been corrected. Thank you for pointing this out.
Best regards,
Reviewer 3 Report
Comments and Suggestions for Authors
This research is certainly interesting. The analysis of the possible relationships between belief/or non-belief in God and the Transcendence and "civic engagement" is a relevant field of research. Above all, investigating the relationship between belief in a transcendent world and civic and political commitment is a fruitful research path.
However, the author should better clarify what is meant by "transcendence". The term itself therefore needs to be better defined. This concept, being particularly complex, allows multiple interpretations. The different interpretations then give rise to very different social beliefs and practices.
The sources on which the research is based are mostly psychosocial. This is fine, especially in research of this type. However, I would suggest strengthening the sociological literature. For example, for the reflection carried out by the author, I would suggest a reference to the work of M. Gauchet (especially the "Disenchantment of the world"). An integration relating to the concepts of "heteronomy" and "autonomy" developed by the French scholar will be sufficient. It would also be very interesting - perhaps in future research - to develop the author's reflection by applying it to the field of new forms of spirituality.
Furthermore, in the sociology of religion, the exceptional importance of prayer as a practice is highlighted. It would be useful to integrate this aspect with a paragraph in which the exceptional nature of this practice is underlined (by providing a reference to sociological literature). Also because prayer plays a key role in the analysis carried out by the author.
Having said this, the research is carried out in a rigorous and clear way, and the hypothesis underlying the work (especially that of the relationship between belief in transcendence and civil and political engagement) is original and worthy of future developments (also in the analysis - it is a suggestion - of new forms of spirituality).
Author Response
Reviewer 3
Answer: We would like to thank you for sharing inspiring comments expanding the perspective of the proposed manuscript.
Comments and Suggestions for Authors
This research is certainly interesting. The analysis of the possible relationships between belief/or non-belief in God and the Transcendence and "civic engagement" is a relevant field of research. Above all, investigating the relationship between belief in a transcendent world and civic and political commitment is a fruitful research path.
However, the author should better clarify what is meant by "transcendence". The term itself therefore needs to be better defined. This concept, being particularly complex, allows multiple interpretations. The different interpretations then give rise to very different social beliefs and practices.
Answer: Thank you for this important suggestion, we have elaborated on the understanding of transcendence in the presented text and pointed out possible broader understandings, in lines 144-150.
The sources on which the research is based are mostly psychosocial. This is fine, especially in research of this type. However, I would suggest strengthening the sociological literature. For example, for the reflection carried out by the author, I would suggest a reference to the work of M. Gauchet (especially the "Disenchantment of the world"). An integration relating to the concepts of "heteronomy" and "autonomy" developed by the French scholar will be sufficient. It would also be very interesting - perhaps in future research - to develop the author's reflection by applying it to the field of new forms of spirituality.
Answer: Works of this philosopher and historian are very interesting and can provide a canvas for interpreting the results, which we tried to suggest, in lines 562-571.
Furthermore, in the sociology of religion, the exceptional importance of prayer as a practice is highlighted. It would be useful to integrate this aspect with a paragraph in which the exceptional nature of this practice is underlined (by providing a reference to sociological literature). Also because prayer plays a key role in the analysis carried out by the author.
Answer: Undoubtedly, this point is very valid, but given the vast sociological literature on the importance of prayer as a practice, we have decided to focus primarily on the psychological perspective. In subsequent analyses, we will try to incorporate the sociological perspective as well.
Having said this, the research is carried out in a rigorous and clear way, and the hypothesis underlying the work (especially that of the relationship between belief in transcendence and civil and political engagement) is original and worthy of future developments (also in the analysis - it is a suggestion - of new forms of spirituality).
Answer: We would like to thank you for this evaluation.
Best regards,